# MSI and EBV Positive Gastric Cancer’s Subgroups and Their Link with Novel Immunotherapy

**DOI:** 10.3390/jcm9051427

**Published:** 2020-05-11

**Authors:** Maria Grazia Rodriquenz, Giandomenico Roviello, Alberto D’Angelo, Daniele Lavacchi, Franco Roviello, Karol Polom

**Affiliations:** 1Division of Medical Oncology, Department of Onco-Hematology, IRCCS-CROB, Referral Cancer Center of Basilicata, via Padre Pio 1, 85028 Rionero, Vulture (PZ), Italy; grazia.rodriquenz@gmail.com; 2Department of Health Sciences, University of Florence, viale Pieraccini, 6, 50139 Florence, Italy; giandomenicoroviello@hotmail.it; 3Department of Biology and Biochemistry, University of Bath, Bath BA2 7AY, UK; ada43@bath.ac.uk; 4Azienda Ospedaliera Careggi University Hospital of Florence and University of Florence, 50134 Florence, Italy; daniele.lavacchi@yahoo.it; 5Department of Medical, Surgical and Neuro Sciences, Section of Surgery, Azienda Ospedaliera Universitaria Senese, University of Siena, 53100 Siena, Italy; roviello@unisi.it; 6Department of Surgical Oncology, Gdansk Medical University, 80-210 Gdansk, Poland

**Keywords:** MSI, EBV, molecular subtypes, immunotherapy

## Abstract

Gastric cancers have been historically classified based on histomorphologic features. The Cancer Genome Atlas network reported the comprehensive identification of genetic alterations associated with gastric cancer, identifying four distinct subtypes— Epstein-Barr virus (EBV)-positive, microsatellite-unstable/instability (MSI), genomically stable and chromosomal instability. In particular, EBV-positive and MSI gastric cancers seem responsive to novel immunotherapies drugs. The aim of this review is to describe MSI and EBV positive gastric cancer’s subgroups and their relationship with novel immunotherapy.

## 1. Introduction 

Gastric cancer (GC), despite its declining incidence, is one of the most common causes of cancer-related mortality worldwide. This malignancy includes a heterogeneous group of neoplastic epithelial lesions with a variety of predisposing conditions and etiological factors. GCs have been historically classified based on histomorphologic features. Lauren classification distinguishes two subtypes of GC (intestinal or diffuse), while the World Health Organisation (WHO) system classifies GC into four subtypes (papillary, tubular, mucinous and poorly cohesive). Although current histopathologic systems can sometimes influence endoscopic or surgical choices, they remain insufficient to reflect the molecular and genetic characteristics of GC and guide surgical and medical strategies in the era of precision medicine [1]. As in many other tumors, molecular drivers might have a crucial role in pathogenetic and therapeutic in GC.

Currently, only two biomarkers are available to predict treatment effectiveness in patients: human epidermal growth factor receptor 2 (HER2) for trastuzumab and programmed death-ligand 1 (PD-L1) expression for pembrolizumab. In the ToGA (Trastuzumab for Gastric Cancer) trial, the addition of trastuzumab to chemotherapy lead to a significant improvement in overall survival (OS) in patients with HER2 overexpressing GCs (13.8 months vs 11 months, respectively; *p* = 0.046) [2]. In September 2017, the Food and Drug Administration (FDA) granted accelerated approval of Pembrolizumab, a PD-1 inhibitor, for the treatment of patients with recurrent locally advanced PD-L1 positive gastric adenocarcinomas that progressed on or after two or more prior systemic therapies. Approval was based on the results of the KEYNOTE-059 study, an open-label, multicenter trial that enrolled 259 patients. PD-L1 positivity was determined on a combined positive score (CPS) ≥ 1. CPS is calculated by the number of PD-L1 staining cells (tumor cells, lymphocytes, macrophages) divided by a total number of tumor cells evaluated, multiplied by 100. Among 143 patients (55%) with tumors expressing PD-L1 and who were either microsatellite stable (MSS) or had unknown microsatellite-unstable/instability (MSI) or DNA mismatch repair deficiency (dMMR) status, the overall response rate (ORR) was 13.3%; 1.4% had complete response (CR) and 11.9% had PR. Among the 19 responding patients, the response duration ranged from 2.8 to 19.4 months, with 11 patients (58%) having response durations of 6 months or longer and 5 patients (26%) having response durations of 12 months or longer. Among the 259 patients enrolled in KEYNOTE-059, 7 (3%) had tumors that were determined to be MSI-H. Responses were observed in 4 of these 7 patients (ORR 57%), with one CR. The response duration ranged from 5.3 to 14.1 months. Even though this evidence determined the approval of trastuzumab and pembrolizumab in advanced GC, the weak molecular selection of patients included in clinical trials is still an issue and may limit evaluation of the benefit of many therapeutic agents, such as antiangiogenic molecules and more recent immunomodulatory agents.

## 2. Molecular Landscape and Classification of Gastric Cancer

Recent progress in genomic technology has now allowed GCs to be studied at the molecular level facilitating the identification of potentially “druggable” alterations in GC, such as gene mutations, chromosomal alterations, transcriptional changes and epigenetic derangements (Table 1).

In 2014, a milestone research carried out by The Cancer Genome Atlas (TCGA) network reported the comprehensive identification of genetic alterations associated with GC, testing 295 frozen GC tissues from untreated patients with six different molecular platforms, including whole exome sequencing (WES), messenger RNA sequencing, microRNA sequencing (miRNA), array-based DNA methylation profiling, reverse-phase protein array and array-based somatic copy number analysis. On the basis of an integrative analysis of this molecular information, the TCGA team identified four distinct subtypes—EBV-positive (8.8%), which displayed recurrent *PIK3CA* mutations, extreme DNA hypermethylation and amplification of *JAK2*, *CD274* (also known as *PD-L1*) and *PDCD1LG2* (also known as *PD-L2*), microsatellite-unstable/instability (MSI, 21.7%), which shows elevated mutation rates, including mutations of genes encoding targetable oncogenic signalling proteins, genomically stable (19.7%), which were enriched for the diffuse histological variant and mutations of *RHOA*, fusions involving RHO-family GTPase-activating proteins, CDH1 somatic mutations, CLDN18–ARHGAP6 or ARHGAP26 fusions and chromosomal instability (CIN, 49.8%), which showed marked aneuploidy and focal amplification of receptor tyrosine kinases. Interestingly, neither racial nor survival differences were found among each subgroup [3].

In the subsequent year, the Asian Cancer Research Group (ACRG) proposed a new molecular classification for GC [4]. On the basis of whole-genome sequencing, gene expression profiling, genome-wide copy number microarrays and targeted gene sequencing, the ACRG first divided the GCs into MSI and microsatellite stable (MSS) types. Then, secondarily, the MSS GC were divided into epithelial-mesenchymal transition (EMT), TP53+ and TP53- groups.

MSS/EMT was characterized by the worst prognosis and a higher chance of recurrence (63%) which occurs mainly in peritoneum; it was predominantly associated with Lauren diffuse histologic type (> 80%), diagnosed at a younger age and clinical stage III/IV. ACRG MSS/EMT subtype had a partial overlap with GS subtype in TCGA. On the contrary, MSI subtype occurred predominantly at the antrum (75%) and over 60% were diagnosed as the intestinal subtype and at an early stage (I/II). Moreover, MSI subgroup showed the best prognosis, followed by MSS/TP53+ and MSS/TP53-. EBV infection occurred more frequently in the MSS/TP53- group than in other groups.

A comparison between TCGA and ACRG data allows distinguishing both similarities and differences. Both TCGA and ACRG classifications identify MSI as a separate subgroup of GC. GS was associated with MSS/EMT, EBV to MSS/TP53+ and CIN to MSS/TP53- but it was unlikely to find a complete correspondence among these other subgroups. While CIN and GS TCGA subtypes tumors were present across all ACRG subtypes, TCGA GS, EBV+ and CIN subtypes were enriched in ACRG MSS/EMT, MSS/TP53+ and TP53− subtypes, respectively.

According to histological features, the majority of TCGA diffuse subtype cases (57%) belonged to GS subgroup but only 27% of the cases were present in the correspondent ACRG MSS/EMT subgroup, suggesting that TCGA diffuse subtype case were less heterogeneous. In TCGA, EBV positive GCs represented a distinctive subgroup, whereas in ACRG cohort EBV infection was found more frequently in MSS/TP53+ subtype, without hypermethylation or hypermutation. PI3K mutations and ARID1A mutations were less prevalent in MSS than in EBV positive subtype.

As previously reported, these two classifications had a partial overlap. Several differences, including geografic differences in the two patient populations (Eastern vs Western patients) and differences in tumor sampling, amplified by the use of distinct platforms, have been considered responsible for the mismatch across categories of these two classifications [5,6]. In a more recent paper, Li et al. proposed a novel classification system, based on the aggregation of somatic molecular profiles of 544 GC patients from previous genomic studies. GCs were divided into regular (86.8%; 2.4 mutations/Mb; range, 0–8.3) and hypermutated (13.2%; 20.5 mutations/Mb; range, 9.6–200.2) subtypes based on mutation burden, the latter of which showed a marked overrepresentation of samples with microsatellite instability. The regular type was subclassified into 2 subgroups, C1 and C2, with distinct clinical outcomes, independently of disease staging. C1 carried mutations in TP53, XIRP2 and APC and correlated with significantly better prognosis, while C2 overexpressed mutations in ARID1A, CDH1, PIK3CA, ERBB2 and RHOA [7]. Interestingly, in this analysis, CDH1 mutations were found as an independent prognostic factor in diffuse-type but not intestinal-type GC regardless of TNM staging. Unfortunately, the application in the clinical setting of these molecular classifications is still limited. The highly complex methodology, lack of prospective validation and short period of follow-up limit the reproducibility in standard laboratories lacking cutting edge technologies. Nevertheless, the great amount of data obtained from these multi-omics profiling classifications has helped to redefine the genetic and pathogenic landscape of GC and to elucidate novel molecular targets and therapeutic strategies paving the way for precision oncology. Moreover, these classifications are expected to improved patient stratification for novel clinical trials. The molecular classification of GCs will probably change the way of thinking in GC oncology. Because of the somewhat promising targeted therapies that are under investigation in clinical trials, EBV positive GCs and MSI GCs seem to have the highest importance [8], in particular we will focus on EBV and MSI subgroups that seem to have the most clinically relevant impact (Figure 1).

## 3. EBV Positive GCs

EBV infection is found in > 90% of world population presenting as lifelong latent infection and Epstein Barr Virus is present in about 9% of all gastric adenocarcinomas worldwide [9]. Along with Helicobacter Pylori (HP) infection EBV assign to GC the primacy of infection-related cancer mortality. The EBV-positive subtype of gastric adenocarcinoma is conventionally identified by in situ hybridization (ISH) for viral nucleic acids but next-generation sequencing represents a potential alternative.

According to TCGA analysis, EBV GC accounts for 8.2% of all GCs and is associated with male predominance (81%), younger age, gastric fundus or body location (62%), with no significant differences in proportion between intestinal and diffuse histology [10].

Tumors positive for EBV display recurrent *PIK3CA* (80%), ARID1A (55%) and BCOR (23%) mutations, extreme CpG hypermethylation (including both promoter and no promoter CpG islands and universal CDKN2A promoter hypermethylation (45%) thaa results in expression of CDKN2A) and JAK2 amplification (25%). Fifteen % of EBV-positive GCs also present amplification of 9p24.1 locus *CD274* (also known as *PD-L1*) and *PDCD1LG2* (also known as *PD-L2*) which result in enhanced neoepitope presentation. All these features suggest a potential role for PI3K inhibitors, JAK2 inhibitors and immune checkpoint inhibitors. In EBV positive GCs PI3KCA mutations are spread across many gene segments and not only in helical or kinase domains and induce the activation of PIK3/mTOR pathway.

Interestingly, the methylation in EBV-positive GCs is more extensive than observed in the MSI GC subgroup and in any tumor, type studied by the TCGA. Moreover, even the pattern of methylation is different, with the EBV-positive GCs showing CDKN2A (p16) promoter hypermethylation but not MLH1 hypermethylation [11]. EBV-positive GCs, in general, lack TP53 mutation, although TP53 was nearly always mutated in “chromosome instability” cancers [3].

In the largest analysis performed by Camargo et al., EBV-GC showed better survival than EBV-negative GC [12]. In a study from Portugal, the EBV GC incidence was 8.4% with statistically significant difference in histological type, upper third position, lower lymph node metastases, with a tendency for better OS [13].

As described in a retrospective cohort of 160 advanced GC patients who underwent potentially curative surgery with or without chemotherapy, phosphorylated AKT positive patients had a good prognosis in terms of OS and relapse-free survival, suggesting that pAKT may be a biomarker for better outcomes for GC patients undergoing gastrectomy regardless of the PIK3CA mutation status [14].

Various PI3K/AKT/mTOR inhibitors have been tested in the metastatic setting. Everolimus, a mTOR inhibitor, has shown potential benefit in phase II trials on advanced GCs [15,16] but did not reach any significant improvement in OS in subsequent phase III trials [17].

In a recent paper published by Chen et al., PI3K/mTOR dual inhibitor BEZ235 displayed higher therapeutic efficiency than everolimus or the MEK inhibitor AZD6244 in paclitaxel-resistant GC cells [18].

Regarding immune system dysfunction, TCGA network observed that EBV-positive cancers are characterized by hyperactive adaptive and innate immunity, showing T-cell activation via the cytokines IL-2, IL-12, IL-23 and IL-27 [3].

EBV positive GCs had a lower mutation burden but stronger evidence of immune infiltration compared with MSI tumors and had higher expression of immune checkpoint pathway (PD-1, CTLA-4 pathway) genes compared with MSS tumors [19].

Recent novel strategies using EBV-directed therapy are being investigated as a potential therapy for different types of EBV-positive tumors. EBV-directed therapy induced the lytic form of EBV to convert infected cells from latent to replicative phases of viral infection, which is hypothesized to trigger cell death with potential for bystander killing of adjacent cells [20].

## 4. MSI Positive GCs

A defective DNA mismatch repair (MMR) system is well known as the leading cause of the high mutational burden found in several gastrointestinal cancers. The threshold above which tumors are considered hypermutated, however, depends on sequencing methods and type of cancer (20.5 mutations/Mb in GCs) [7]. MMR deficiency resulting from mutational inactivation or epigenetic silencing of DNA mismatch repair genes (e.g., MSH2, MSH3, MSH6, MLH1 and PMS2) causes MSI, which is characterized by alteration in the length within short repeated DNA sequences (microsatellites) [21].

MSI occurs in a sizable share of GC (8%–37%), in which MSI-H phenotype is mostly derived from epigenetic hypermethylation of MLH1 rather than germline mutation [22]. According to the TCGA molecular classification, MSI occurs in 22% of GCs.

MSI-H GCs show peculiar clinical and molecular features and are usually associated with female sex, older age, distal location, no lymph node involvement, intestinal Lauren histotype, lower local invasion capacity, earlier stage and better survival [23]. They are often diagnosed at clinical stage I/II with the best prognosis. MSI in GCs seems to be a positive prognostic factor [24] and recurrence rate after surgical resection of primary MSI positive GCs is the lowest among all four subtypes [3] (22%).

In a post-hoc analysis of CLASSIC trial, MSI-H was an independent prognostic factor for disease-free survival (DFS). In addition, 5-year DFS was significantly higher in patients with MSS tumors who received adjuvant chemotherapy compared with those who underwent surgery alone (66.8% vs 54.1%; *p* = 0.002). In contrast, no benefit from adjuvant chemotherapy was observed in patients with MSI-H tumors (83.9% vs 85.7%; *p* = 0.931) [25]. 

Similar results were obtained from a post-hoc analysis of the MAGIC trial. Among patients who underwent surgery alone, median OS was not reached in the dMMR/MSI-H group vs 20.5 months in the non-dMMR/MSI-H group (*p* = 0.09). In contrast, among patients who received perioperative chemotherapy, median OS was 9.6 months in the dMMR/MSI-H group vs 19.9 months in the non-dMMR/MSI-H group (*p* = 0.03) [26].

Also a recent meta-analysis including four trials (i.e., MAGIC, CLASSIC, ARTIST and ITACA-S) showed higher survival rates in patients with MSI tumors compared to those with MSS tumors. No benefit was observed in MSI patients from the addition of chemotherapy to surgery [27]. In a large prospectively surgical database, patients with MSI-H GCs may have long term survival despite R+ marginal status after surgical resection [28].

Frequent chromosome 8 gain occurs in MSI subtype, whereas 18q loss prevails in EBV-positive GC [29].

Mutational analysis of MSI-H GCs in TCGA research identified 37 genes significantly mutated, including TP53, KRAS, PI3K, ARID1A, PTEN, ERBB2, ERBB3. Because ARID1A is frequently mutated in both EBV and MSI subtypes, its mutation alone is not likely to constitute an alternative GC pathway. Common alterations in major histocompatibility complex (MHC), including B2M and MHC-B, often occurred suggesting these events benefit hypermutated tumors by reducing antigen presentation to the immune system. Although MSI cases generally lacked targetable amplifications, mutations in *PIK3CA*, *ERBB3*, *ERBB2* and *EGFR* were noted, with many mutations at ‘hotspot’ sites seen in other cancers. Absent from MSI GCs were *BRAF* V600E mutations, commonly seen in MSI colorectal cancer [30]. Genes in the TGF-β pathway (e.g., TGFBR2, ACVR2A, SMAD4 and ELF3) predicted to be key drivers in MSI are frequently mutated in this subgroup, suggesting an important role in GC biology.

The particularly high rate of somatic mutations in these tumors promotes the generation of neo-antigens capable of eliciting an immune response, making MSI-H tumors suitable for immune checkpoint blockade therapy.

## 5. Immunotherapy in EBV Positive and MSI-H GCs

The link between infection, chronic inflammation and malignancy recognized in GC suggests that targeting the immune system may lead to improved outcomes in this type of tumors which are constitutively resistant to systemic treatments [31].

Considering the molecular pattern of each subgroup in TCGA classification, it is likely that the majority of patients responding to single agent checkpoint inhibitor may belong to EBV and MSI. Alternatively, patients with genomically stable subtype and chromosomally unstable subtype may need combination immunotherapy.

Several attempts have been made to identify predictive markers of immunotherapy response, including a high tumor antigen load, changes in immune-regulatory cytokines and levels of coinhibitory proteins [32]. 

Among predictive factors of immune-checkpoint inhibitor efficacy, genomic aberrations that contribute to the enhanced PD-L1 expression have been demonstrated in both the MSI and EBV subtypes of GCs. PD-L1 expression was observed in approximatively 50% and 94% of tumor cells and immune cells in the EBV subtype and in approximatively 33% and 45% of tumor cells and immune cells in MSI-H tumors [33].

As reported in other types of cancer, negative immune checkpoint proteins have been shown to be upregulated in tumors with a T-cell inflamed phenotype [34]. As consequence, the efficacy of anti-PD-1/PD-L1 agents would not seem to be limited only to MSI and EBV-positive GC subtypes but also against tumors with high degree of lymphocyte infiltration.

Among MMR-deficient tumors, the strong expression of crucial immune checkpoint ligands (e.g., PD-1/PD-L1, LAG-3, IDO and CTLA4) could confer a reduced sesitivity to immunotherapeutic agents [35]. Otherwise, EBV+ GCs demonstrated amplification of genes which encode PD-L1 and PD-L2 and result in enhanced neoepitope presentation.

Several phase II and phase III data show that unselected patients with metastatic GC have response rates of approximately 10%–17%, whereas patients who have PD-L1+ tumors (> 1% of cells) have response rates of 22–27% [36]. Novel combinations or settings are ongoing and definitive data are awaited [37,38].

The ATTRACTION-2 trial is the first phase III immunotherapy trial to demonstrate an improved OS for nivolumab compared with placebo in patients with heavily pretreated metastatic GCs and gastroesophageal cancers. In this randomised, double-blind, placebo-controlled trial they enrolled a total of 493 patients to receive nivolumab (n = 330) or placebo (n = 163) at 49 clinical sites in Taiwan, South Korea and Japan. Using an interactive digital response system, patients were randomly assigned (2:1) to receive intravenously 3 mg/kg placebo or nivolumab every 2 weeks, stratified by number of organs with metastases, ECOG performance status and country. The median OS, the designated primary endpoint was 5.26 months (95% CI 4.60–6.37) in the nivolumab group and 4.14 months (3.42–4.86) in the placebo group (HR 0.63, 95% CI 0.51–0.78; *p* < 0.0001) as the 12-month OS rates were 26.2% (95% CI 20.7–32.0) with nivolumab and 10.9% (6.2–17.0) with placebo. The study reported an ORRof 11.2% with nivolumab versus 0% with placebo and a median duration of response to nivolumab of 9.53 months.Grade 3 and 4 treatment-related events (mainly pruritus, diarrhoea, rash and fatigue) occurred in 34 (10%) of 330 patients in the nivolumab arm and 7 (4%) of 161 patients in the placebo arm [39]. Despite the absence of data about the quality of life, according to this trial nivolumab might be considered a valid new therapy for heavily pre-treated patients affected by gastric or gastro-oesophageal junction cancer.

In September 2017, FDA approved pembrolizumab with accelerated process for patients with advanced GC after at least two previous lines of chemotherapy that expresses PD-L1; PD-L1 expression must be determined by an FDA approved test and have a CPS ≥ 1.

The approval was based on results from KEYNOTE-059 trial, an open-label, 3-cohort, phase II and no randomized trial which enrolled 259 patients diagnosed with heavily pre-treated gastric or gastro-oesophageal junction cancer performed at 67 different hospitals of 17 different countries. After been assessed for human epidermal growth factor receptor 2 (HER2)/neu-negative or HER2/neu positive if previously treated with trastuzumab, PD-L1-positive and DNA mismatch repair tumor status [40], patients received 200 mg of pembrolizumab intravenously every 3 weeks. Overal response rate (ORR), designated as primary endpoint, was 11.6% (95% CI, 8.0%–16.1%; 30 of 259 patients) with CR in 2.3% (95% CI, 0.9%–5.0%; 6 of 259 patients). In PD-L1 positive group, ORR was 15.5% (95% CI, 10.1%–22.4%; 23 of 148 patients) whereas in PD-L1 negative group ORR was 6.4% (95 CI, 2.6%–12.8%; 7 of 109 patients). Among all 174 out of 259 (67.2%) tested patients assessed for MSI, 7 (4.0%) presented MSI-H samples, of these 4 (57.1%) experienced objective response (95% CI, 18.4%–90.1%). Conversely, ORR was lower in non–MSI-H samples (9%, 95% CI, 5.1%–14.4%). In addition, the 18-gene T-cell-inflamed gene expression profiling showed that patients who respond to pembrolizumab had a higher score compared to non-responders; a higher gene expression profiling score was remarkably associated with increased progression-free survival (PFS) and propensity for a response. Grade 3 to 5 treatment-related adverse events occurred in 46 patients (17.8%) mainly characterised by fatigue (18.9%), pruritus (8.9%) and rash (8.5%). This study demonstrated the promising activity and manageable safety of pembrolizumab in heavily pre-treated advanced gastric or gastro-oesophageal junction patients.

The other cornerstone in pembrolizumab approval has been the KEYNOTE-061 trial [41], an open-label, randomised and controlled phase 3 trial that compared pembrolizumab with paclitaxel in patients with advanced gastric or gastro-oesophageal junction cancer that progressed on first-line chemotherapy with platinum and fluoropyrimidine. This study was performed over 148 medical centres in 30 countries enrolling 592 who were randomised (1:1) in blocks of four per stratum with an interactive voice-response and integrated web-response system to receive either standard-dose of paclitaxel (n = 199) or pembrolizumab 200 mg (n = 196) every 3 weeks for up to 2 years. OS and PFS in patients with a PD-L1 CPS of 1 or higher were the primary endpoints. The study showed a median OS of 9.1 months (95% CI 6.2–10.7) for the pembrolizumab group and 8.3 months (7.6–9.0) for the paclitaxel group (HR 0.82, 95% CI 0.66–1.03; one-sided p = 0.0421) as the median PFS was 1.5 months (95% CI 1.4–2.0) in pembrolizumab arm and 4.1 months (3.1–4.2) in paclitaxel arm (HR 1.27, 95% CI 1.03–1.57). Although the better safety profile of pembrolizumab than paclitaxel–treatment-related adverse events occurred in 42 (14%) of the 294 patients who received pembrolizumab and 96 (35%) of the 276 patients who received paclitaxel - pembrolizumab did not decisively increased OS compared to paclitaxel as second-line therapy for advanced gastric or gastro-oesophageal junction cancer with PD-L1 CPS of 1 or higher.

In a multicohort phase II trial, KEYNOTE-158, pembrolizumab was administered to pretreated patients with dMMR/MSI-H tumors of various types, 24 of whom had a GC. Among MSI-H GC patients, ORR was 45.8% (CR in 4 patients) and PFS was 11.0 months. This trial confirmed a remarkable activity of pembrolizumab in patients who have a GC refractory to standard treatments and, therefore, poor prognosis [42].

## 6. How to Better Select Patients for Immunotherapy: Future Directions

Over the last years, the oncology landscape has been revolutionised by immunotherapy which has proven efficacy in several cancers, by helpfully blocking immune checkpoints [43]. Even though several immunotherapy-based trials have reported a wide range of tumor response rate in patients with GC, many phase III trials of targeting agents failed to show a significant survival benefit. (Table 2) Therefore, the need to unveil novel biomarkers to better select patients who might benefit from immunotherapy is warranted. GC has been molecularly characterized by several molecular classifications proposed, as discussed above and different molecular subgroups identified [2,3,4,5,6,7]. However, scientific community agrees that next clinical trials of immune and targeted therapy in the treatment of advanced GC should be tailored based on genomic differences and immunological features in order to achieve a severe impact on treatment responses and clinical outcomes. Japan has recently approved FoundationeOne CDx and NCC Oncopanel, two multiplex gene panels for better genomic profiling of patients affected by advanced GC. Furthermore, other studies suggest that the emerging landscape of circulating tumor DNA (ctDNA) profile should be evaluated before and after the administration of molecular targeted therapies [44,45]. Given the limitations of the invasive biopsies and the biomarker heterogeneity in patients with GC, we speculate that ctDNA might be an efficient tool to better select patients for immunotherapy. A baseline evaluation of the ctDNA might efficiently reflect the mutational load in the tumor and its early variation predict the immunotherapy efficacy after treatment start. As reported by Garlan et al. [46] and Kim et al. [47], the mutational tumor load assessed by ctDNA at baseline was predictive of response to treatment and its decrease at 6 weeks after drug administration was associated with increased Progression free survival (PFS). and tumor response.

Lastly, the addition of the molecular classification of GC using Next Generation Sequence technology provides a powerful tool to the traditional histopathologic classification. These technologies (Table 3) are extremely promising in identifying novel biomarkers that might predict response to therapy, likelihood of relapse or metastasis and tumor behaviour in general [48]. Indeed, the combination of genomic data with traditional clinical staging, including histopathologic variables and TNM, has the power to provide more effective and individualized treatment options. Although the whole genome sequencing is technically practicable and is becoming more and more common in the research daily practice, targeted sequencing and whole exome sequencing (WES) are more employed in clinical practice routinely. Moreover, in gastric and oesophagal cancer several genetic alterations have been reported [6,49,50] (Table 4) and there is an open competition to identify which of these mutations could be targetable.

In conclusion, a significant growth in the knowledge of GC biology has translated into the development of a variety of novel agents that are beginning to show clinical benefit in GCs. In particular, MSI-H and EBV positive subgroups seem more able to respond to novel immunotherapy drugs compared to other molecular subgroups. Therefore, in the near future, pre-clinical and clinical studies are awaited to confirm the effective role of novel immunotherapy in MSI-H and EBV GC patients and to discover biological predictive biomarkers for these subgroups, in order to define the best treatment strategy tailored to the individual patient.

## Figures and Tables

**Figure 1 jcm-09-01427-f001:**
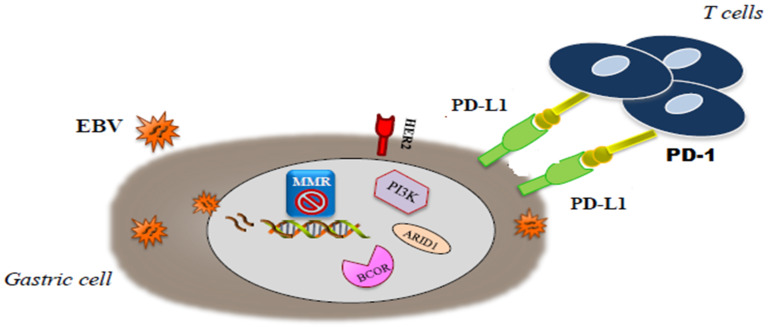
Molecular interaction in microsatellite-unstable/instability (MSI) end. Epstein-Barr virus (EBV) gastric cancer (GC).

**Table 1 jcm-09-01427-t001:** Molecular classification of Gastric Cancer according TCGA (The Cancer Genome Atlas)emt; ACRG (Asian Cancer Research Group) and Li et al. Classification.

	SUBTYPES	MOLECULAR FEATURES
**TCGA (The Cancer Genome Atlas)**	EBV (9%)	- DNA hypermethylation, including CDKN2A (p16) but not MLH1 promoters
		- PIK3CA mutations
		- JAK2 gene amplification
		- PDL1/PDL2 overexpression
	MSI (22%)	- high mutation rate
		- DNA methylation with epigenetic silencing of MLH1
		- Hypermutation of many genes including HLA class 1 factors
	GS (20%)	- molecular alterations in cell adhesion/ cell migration pathways
		- ARID1 and BCOR mutations
	CIN (50%)	- chromosomal instability (CIN)
		- amplification of genes (most encoding tyrosine kinase receptors)
**ACRG (Asian Cancer Research Group)**	MSS/TP53 + (26%)	- frequent EBV positivity
		- intermediate mutation rate
	MSI (23%)	- high mutation rate
	EMT (15%)	- low mutation rate
		- loss of epithelial markers
	MSS/TP53- (36%)	- TP53 mutations
		- genomic instability
**Li et al. Classification**	REGULAR C1	- 2.4 mutations /megabase; range, 0–8.3
		- TP53, XIRP2, APC mutations
	REGULAR C2	- 2.4 mutations /megabase; range, 0–8.3
		- ARID1A, CDH1, PIK3CA, ERBB2, RHOA mutations
		-
	HYPERMUTATED	- 20.5 mutations/megabase; range, 9.6–200.2)

**Table 2 jcm-09-01427-t002:** Latest phase 3 target agents in gastric cancer.

Trial/Author	Target	Agent	Line	Control	Endpoint	Result	Difference mOS (m) (HR)
Keynote061	PD1	Pembrolizumab	2nd	Paclitaxel	OS	Negative	+0.8 (HR 0.82)
JAVELIN300	PD1	Avelumab	3rd	Irinotecan/taxanes	OS	Negative	−0.4 (HR 1.1)
ATTRACTION-2	PD1	Nivolumab	3rd	Placebo	OS	Positive	+1.2 (HR 0.63)
AVAGAST	VEGF-A	Bevacizumab	1st	Placebo (+chemo)	OS	Negative	+2 (HR 0.87)
RAINFALL	VEGFR2	Ramucirumab	1st	Placebo (+chemo)	OS	Negative	+0.4 (HR 0.96)
REGARD	VEGFR2	Ramucirumab	2nd	Placebo	OS	Positive	+1.4 (HR 0.776)
RAINBOW	VEGFR2	Ramucirumab	2nd	Placebo (+chemo)	OS	Positive	+2.2 (HR 0.807)
Li. et al.	VEGFR2	Apatinib	3rd	Placebo	OS	Positive	+1.8 (HR 0.71)
REAL-3	EGFR	Panitumumab	1st	(+chemo)	OS	Negative	−2.5 (HR 1.37)
EXPAND	EGFR	Cetuximab	1st	Placebo (+chemo)	PFS	Negative	−1.3 (HR 1.0)
ToGA	HER2	Trastuzumab	1st	(+chemo)	OS	Positive	+2.7 (HR 0.74)
Logic	HER2	Lapatinib	1st	Placebo (+chemo)	OS	Negative	+1.7 (HR 0.91)
JACOB	HER2	Pertuzumab	1st	Placebo (+chemo + Tmab)	OS	Negative	+3.3 (HR 0.84)
TyTAN	HER2	Lapatinib	2nd	(+chemo)	OS	Negative	+3 (HR 0.84)
GATSBY	HER2	T-DM1	2nd	Taxanes	OS	Negative	−0.7 (HR 1.15)
GRANITE-1	mTOR	Everolimus	2nd/3rd	Placebo	OS	Negative	+1.05 (HR 0.9)
GRANITE-2	mTOR	Everolimus	2nd	Placebo (+chemo)	OS	Negative	+1.0 (HR 0.92)
RILOMET1	HGF	Rilotumumab	1st	Placebo (+chemo)	OS	Negative	−2.9 (HR 1.36)
METgastric	MET	Onartuzumab	1st	Placebo (+chemo)	OS	Negative	−0.3 (HR 0.82)
GOLD	PARP	Olaparib	2nd	Placebo (+chemo)	OS	Negative	+1.9 (HR 0.79)
BRIGHTER	STAT3	Napabucasin	2nd	Placebo (+chemo)	OS	Negative	−0.4 (HR 1.01)

**Table 3 jcm-09-01427-t003:** Next-generation sequencing terms.

Next-Generation Sequencing Techniques	
Whole-genome sequencing (WGS)	Single-nucleotide resolution of all genome bases
Whole-exome sequencing (WES)	Single-nucleotide resolution of protein-codon areas of the genome
Targeted sequencing	Covers limited subsets of candidate genes
RNA sequencing	Sequencing of each RNA transcript
Gene expression profiling	Evaluates the RNA level of a single gene with further functional associations; cell environment as potential bias

**Table 4 jcm-09-01427-t004:** Most recurrent genetic alterations in oesophagal and gastroesophageal cancers.

Esophageal Cancer		Gastroesophageal Cancer	
Gene	Frequency (%)	Gene	Frequency (%)
TP53	60–93	TP53	14–59
CCND1	33–46	PIK3CA	7–36
CDKN2A	12–47	CDH1	4–36
KMT2D	19–26	HER2	2–32
FAT1	14–27	ARID1A	8–27
KRAS	5–27	KRAS	0–27
EGFR	6–24	PTEN	0–27
NOTCH	9–19	RHOA	0–23
PIK3CA	4–10	APC	3–14
		ERBB3	0–10
		CTNNB1	2–9
		MET	0–9
		SMAD4	4–6
		FBXW7	2–6

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
