# Peer review of "MSI and EBV Positive Gastric Cancer’s Subgroups and Their Link with Novel Immunotherapy"

_jcm, 2020, doi:10.3390/jcm9051427_

Round 1

Reviewer 1 Report

This is a very nice review summarizing recent advances on molecular subtypes in gastric cancers and their clinical outcome. The authors primarily focus on the EBV and MSI subtypes and the manuscript and figure/tables are well organized.

There are several minor points that should be addressed before publication.

  1. 1. Line 65-68. Genome stable type has RhoA and Rho family mutations, but it also has frequent mutation in CDH1 and TP53. This point should be added here.
  2. 2. In some places, the authors describe “TGCA”, but it should be “TCGA”. Please correct throughout manuscript.
  3. 3. Line 76-93. The correlation between Lauren diffuse type and MSS/EMT and GS type is unclear. Please make this point clearer in the text, possibly with additional graphical schema.
  4. 4. In Fig.1, both EBV and MSI cancer cells appear to express PDL1. Is there any evidence supporting that MSI cancer cells rather than immune cells overexpress PDL1? Please clarify and if necessary please revise the figure.
  5. 5. Line 120-121. EBV is infected in >90% world population but EBV is positive only 9% of gastric cancer cases. Is there any reason why only small subset of EBV infectants develop gastric cancer?
  6. 6. Line 129-130. What is the frequency of CDKN2A hypermethylation and JAK2 amplification? Does CDKN2A hypermethylation lead to reduced expression of CDKN2A?
  7. 7. Line 148-150. Is there any subgroup analysis focusing on the effect on EBV+ cancers by PI3K inhibitors?
  8. 8. Line 155. Typo IL-23 e IL-27.
  9. 9. There is a review describing similar molecular aspects in gastric cancers (Hayakawa Y, Nat Rev Cancer. 2016 Apr 26;16(5):305-18.), this work should be cited.

Author Response

  1. Line 65-68. Genome stable type has RhoA and Rho family mutations, but it also has frequent mutation in CDH1 and TP53. This point should be added here.

We changed the sentence as follow:

“genomically stable (19.7%), which were enriched for the diffuse histological variant and mutations of RHOA, fusions involving RHO-family GTPase-activating proteins, CDH1 somatic mutations, CLDN18–ARHGAP6 or ARHGAP26 fusions”

  1. In some places, the authors describe “TGCA”, but it should be “TCGA”. Please correct throughout manuscript.

We have corrected the term “TCGA” throughout manuscript.

  1. Line 76-93. The correlation between Lauren diffuse type and MSS/EMT and GS type is unclear. Please make this point clearer in the text, possibly with additional graphical schema.

As suggested, we reformulated the following sentences to better specify the correlation between Lauren diffuse type and MSS/EMT and GS type:

MSS/EMT subtype was characterized by the worst prognosis and a higher chance of recurrence (63%) which occurs mainly in peritoneum; it was predominantly associated with Lauren diffuse histologic type (> 80%), diagnosed at a younger age and clinical stage III/IV. ACRG MSS/EMT subtype had a partial overlap with GS subtype in TCGA

  1. In Fig.1, both EBV and MSI cancer cells appear to express PDL1. Is there any evidence supporting that MSI cancer cells rather than immune cells overexpress PDL1? Please clarify and if necessary please revise the figure.

Thank you for this comment, the figure is only to report the topic of the manuscript in one figure and it would not suggest that there is evidence supporting that MSI cancer cells rather than immune cells overexpress PDL1.

  1. Line 120-121. EBV is infected in >90% world population but EBV is positive only 9% of gastric cancer cases. Is there any reason why only small subset of EBV infectants develop gastric cancer?

The mechanism by which EBV infects human B lymphocytes is quite clear. In contrast, the mechanism by which EBV infects human epithelial cells remains unclear. Overall, the precise role of EBV in the development of gastric cancer is not fully understood, as unclear is the exact mechanism by which EBV promotes epithelial cell growth. Methylation of promoter region in APC, p16, MINT1, MLH1, TP73, and HOXA10 and down regulation of CXXC4 and TIMP2 were identified in EBV-positive gastric cancer. However, the evidence is too limited to deal with it in a summary way.

  1. Line 129-130. What is the frequency of CDKN2A hypermethylation and JAK2 amplification? Does CDKN2A hypermethylation lead to reduced expression of CDKN2A?

According your suggestion we include the data on CDKN2A hypermethylation and JAK2 amplification along the manuscript, and we added one sentence on the reduced expression of CDKN2A for its gene methylation.

  1. Line 148-150. Is there any subgroup analysis focusing on the effect on EBV+ cancers by PI3K inhibitors?

Thank you for this comment, I have found no relevant data on the use of by PI3K inhibitors in EBV+ cancers.

  1. Line 155. Typo IL-23 e IL-27.

Corrected as requested

  1. There is a review describing similar molecular aspects in gastric cancers (Hayakawa Y, Nat Rev Cancer. 2016 Apr 26;16(5):305-18.), this work should be cited.

As suggested, we added the following reference:

Hayakawa Y1, Sethi N2, Sepulveda AR, et al. Oesophageal adenocarcinoma and gastric cancer: should we mind the gap? Nat Rev Cancer. 2016;16(5):305-18.

Reviewer 2 Report

The manuscript by Rodriquenz et al is a review article on role of immunotherapy in MSI and EBV positive gastric cancers. The manuscript is well written but not suitable for publication in the current form. My specific comments are as follows:

1.  Certain sections of the manuscript has been copied from other published articles, without adequate attribution (referencing). Some examples are as follows:

  • Page 3, line 94-97: has been borrowed from article by Lionel Kankeu Fonkoua et al, published in Biomedicine 2018; 6(1),32.
  • Several sentences on page 6, under section titled 'Immunotherapy in EBV positive and MSI-H GCs', have been taken from the previously published article by Ronan Kelly, titled 'Immunotherapy in Esophageal and Gastric Cancers', in the 2017 ASCO Educational Book.

Consider paraphrasing such sections so as to avoid plagiarism, and provide references wherever applicable.

Author Response

  1. Certain sections of the manuscript has been copied from other published articles, without adequate attribution (referencing). Some examples are as follows:

Page 3, line 94-97: has been borrowed from article by Lionel Kankeu Fonkoua et al, published in Biomedicine 2018; 6(1),32.

We have added the specific reference and reworded the sentences as follow:

As previously reported, these two classifications had a partial overlap. Several differences, including geografic differences in the two patient populations (Eastern vs Western patients) and differences in tumor sampling, amplified by the use of distinct platforms, have been considered responsible for the mismatch across categories of these two classifications.

REF Kankeu Fonkoua L, Yee NS. Molecular Characterization of Gastric Carcinoma: Therapeutic Implications for Biomarkers and Targets. Biomedicines. 2018;6(1):32.

Several sentences on page 6, under section titled 'Immunotherapy in EBV positive and MSI-H GCs', have been taken from the previously published article by Ronan Kelly, titled 'Immunotherapy in Esophageal and Gastric Cancers', in the 2017 ASCO Educational Book.

Consider paraphrasing such sections so as to avoid plagiarism, and provide references wherever applicable.

As suggested, we revised page 6 (section titled 'Immunotherapy in EBV positive and MSI-H GCs) with the following sentences

Several attempts have been made to identify predictive markers of immunotherapy response, including a high tumor antigen load, changes in immune-regulatory cytokines, and levels of coinhibitory proteins. 33

Among predictive factors of immune-checkpoint inhibitor efficacy, genomic aberrations that contribute to the enhanced PD-L1 expression have been demonstrated in both the MSI and EBV subtypes of GCs. PD-L1 expression was observed in approximatively 50% and 94% of tumor cells and immune cells in the EBV subtype and in approximatively 33% and 45% of tumor cells and immune cells in MSI-H tumors34.

As reported in other types of cancer, negative immune checkpoint proteins have been shown to be upregulated in tumors with a T-cell inflamed phenotype35. As consequence, the efficacy of anti-PD-1/PD-L1 agents would not seem to be limited only to MSI and EBV-positive GC subtypes, but also against tumors with high degree of lymphocyte infiltration.

Among MMR-deficient tumors, the strong expression of crucial immune checkpoint ligands (e.g. PD-1/PD-L1, LAG-3, IDO and CTLA4) could confer a reduced sesitivity to immunotherapeutic agents36.

Kelly RJ. Immunotherapy for Esophageal and Gastric Cancer. American Society of Clinical Oncology Educational Book 37 (2018) 292-300.

Reviewer 3 Report

Very good. Please  check abbrevations carefully and add some text to table legends. It is nice if the reader can understand the tables without going through the text.

Author Response

Very good. Please check abbreviations carefully and add some text to table legends. It is nice if the reader can understand the tables without going through the text.

Thank you for this comment, we have included abbreviations and text to table legends.

Round 2

Reviewer 2 Report

Authors have made changes to the manuscript based on reviewers' comments, which are appropriate.